# From Intersex Activism to Law-Making—The Legal Ban of Intersex Genital Mutilation (IGM) in Greece

**Nikoletta Pikramenou**

Faculty of Sociology, University of Warsaw, 00-927 Warsaw, Poland; n.pikramenou@uw.edu.pl

**Abstract:** In 2022, Greece became the fifth country in the world to legally ban Intersex Genital Mutilation (IGM). The bill was prepared by the Ministry of Health and the intersex-led organisation "Intersex Greece". Even though the organisation was only established in 2021, it was actively engaged in the whole law-making process, which resulted in a legal text that became a best practice worldwide. This article tracks the history of the intersex movement in Greece and shows that the movement emerged around 2009. Then, based on online interviews, blogs, videos and articles, all strategies and alliances used by the movement over the years to advocate for intersex rights are explored, especially in the year 2017 when the law on Legal Gender Recognition (LGR) was passed and in 2022 when IGM was banned. Furthermore, online public documents from the Greek Parliament are consulted to provide a comprehensive analysis of how the social, cultural, economic, and political environment in the country affected these legal developments. Based on the above evidence, this article shows that the law-making process on IGM in Greece started 13 years before the law and was the outcome of a long process of multiple and unique intersecting factors.

**Keywords:** intersex movement; legal change; IGM ban; Intersex Greece

## 1. Introduction

Intersex people are born with sex characteristics that do not fit typical binary notions of male or female bodies.[1] As of 2019, 131 million people have been born with intersex traits (Ibid. See also (Ghattas 2019, p. 9)), meaning that almost 1 person in 60 has a variation of sex characteristics.[2] Even though the term "intersex" is an umbrella term used to describe a wide range of natural bodily variations, the term "middlesex" in Greece (μεσοφυλικός/mesofylikos or μεσόφυλος/mesofylos) is often wrongly used, failing to express the intersex reality. Thus, the intersex community has been advocating to completely remove it from public documents and replace it with the term "intersex" written in Greek letters (ίντερσεξ). This term might not be Greek, but it is preferred because, in the Greek language, the term for trans people is "diemfylikoi" (διεμφυλικοί) and is confused with the term for intersex, which is "diafylikoi" (διαφυλικοί) (Intersex Greece 2023, p. 8).

According to Intersex Greece, the sole intersex-led organisation in the country, the initiation of the intersex movement's organisational efforts dates back to 2006, when a group of intersex women and parents of intersex girls created a collective. In 2013, a secret Facebook group of mothers of XXY children started its activity online. The group initially used medical names such as "Klinefelter syndrome" and "Turner syndrome", as its members were not aware that all these variations were grouped under the umbrella term "intersex" (Ibid.). In 2021, this group evolved into an organisation, and on 19 July 2022, Greece became the fifth country in the world to ban Intersex Genital Mutilation (IGM)[3] and criminalise doctors who perform it with Law 4958/2022 (Intersex Greece 2022). Preceding Greece, Malta, Portugal, Germany, and Iceland had already prohibited Intersex Genital Mutilation (IGM), with only Malta going a step further by criminalising it (see Section 3).

The primary objective of this article is to demonstrate the intrinsic connection between the legal changes that transpired in 2022 and the organisational efforts within the intersex

community—a connection influenced by a variety of cultural, social, political, and economic factors. In pursuit of this objective, the article traces the evolution of the intersex movement in Greece by examining available online evidence. The investigation reveals that the Greek intersex movement began gaining visibility around 2009, marked by the publication of an online blog wherein an intersex individual shared their personal story. Then, based on online interviews, blogs, videos, and articles by intersex activists, all strategies and alliances used by the movement over the years to advocate for intersex rights are explored, especially in 2017, when the law on Legal Gender Recognition (LGR) was passed, and 2022, when IGM was banned. Furthermore, online journal articles and public documents from the Greek Parliament are consulted to conduct a comprehensive analysis of how the social, cultural, economic, and political environment in the country affected these legal developments. This study also incorporates sources from countries other than Greece to underscore the linkages between national developments and broader European and global contexts.

In this article, an online methodology was preferred as intersex organisations have flagged the risks that traditional methods of collecting and analysing data through interviews held by non-intersex researchers may entail. For instance, Morgan Carpenter mentioned in 2012 that "research on intersex populations frequently suffers from framing effects" and highlighted the importance of community-based participatory research that recognises and responds to those distinct concerns (Carpenter 2012). Nonetheless, in 2021, Intersex Greece publicly denounced a university research project where its members participated and were interviewed by non-intersex researchers. The organisation claimed that the analysis and interpretation of data were problematic, leading to an "incomplete and non-inclusive portrayal of the intersex reality and intersex experience" (Intersex Greece 2023, p. 28). This paper seeks to deploy the plethora of already published online testimonies and interviews by intersex individuals and offer an alternative method, in an effort not to re-traumatise a community already burdened with physical and psychological distress. The author used sources that are online and public and produced by key figures of the debate following purposive sampling, led by a historical analysis of the sources available. The search engine that was used was Google, and 15 online pages are included in this article based on the selection criteria.

Lastly, this article aims to expand the field of intersex studies as, until now, it has mostly focused on Catholic, Protestant, Western, Northern European, and Anglo-Saxon countries,[4] failing to depict the situation of intersex rights in countries such as Greece, which is an orthodox country, situated in Southeastern Europe on the southern part of the Balkan Peninsula. This occurrence is attributed to various factors, such as the nascent nature of the field of intersex studies, economic disparities among countries, and language barriers. To address the language barrier, two annexes have been appended to the conclusion of this article, featuring translations of pivotal documents from Greek to English. This inclusion aims to facilitate global dissemination and be useful to individuals actively involved in advocating for intersex rights. It is important to acknowledge that dialogues documented in parliamentary proceedings may potentially be triggering for certain audiences.

## 2. The Emergence of the Greek Intersex Movement

### 2.1. The Very Beginning: Intersex Stories "Blossoming" Online (2009–2017)

The global intersex movement gained momentum in the 1990s (Greenberg 2012, p. 85) and has experienced substantial expansion in the subsequent years, persisting to the present day. According to the Astraea Lesbian Foundation for Justice, the first organisations emerged in Australia, North America, Germany, New Zealand, Argentina, and South Africa (Astraea Lesbian Foundation for Justice 2016, p. 10). During the 2000s, the number of groups working for the human rights of intersex people kept on growing steadily. Between 2012 and 2014 in particular, intersex activists founded more than 10 new groups, including the creation of OII Europe,[5] an umbrella organisation campaigning at the European level (Ibid.). Amidst this explosive development of the intersex movement,[6] the blog "Intersex

Flower Greece" was created in 2009 by a person who identified as an intersex woman born with XY chromosomes and used the online name "Intersex Flower."[7] The year of its creation is not a simple coincidence: in their first post, "Caster Semenya," they explain that they were inspired by the global attention that the athlete received the same year.

Semenya won the women's 800 m gold medal at the World Athletics Championships in Berlin in 2009, and her performance triggered a number of questions (Swarr et al. 2009, p. 657) related to her sex characteristics.[8] Scholars have drawn from Semenya's case to criticise outdated policies that rigidly define the female–male binary as an absolute norm (See, for instance, (De Marcilla Musté 2022)) and suggest education on intersex issues as a tool towards equality (See Swarr et al. 2009; Jensen et al. 2022). Nonetheless, an element that is overlooked in academic discourse is the timing: Semenya's case became viral in 2009, meaning that it coincided with the explosive expansion of the intersex movement (Howe et al. 2017, p. 12). It could be assumed that Semenya's case played a pivotal role in elevating intersex visibility and empowering individuals within the intersex community to claim their rights. This, in turn, may have contributed to the hastening of community organising efforts.

The first post of Intersex Flower received nine responses, sparking prompt discussions on the appropriate terminology to be employed in Greek discourse concerning intersex issues. "Hermaphrodite" was already considered as a term that fails to depict the variations of sex characteristics of intersex people, while the earlier mentioned term "diaphylikos" (διαφυλικός, intersex) was often confused with the term "diemphylikos" (διεμφυλικός, trans). Intersex Flower stated that they had been in touch with intersex women in Greece online. However, organising and publicly disclosing their identity posed challenges due to media perpetuating negative stereotypes about intersex individuals. This was compounded by journalists' lack of awareness regarding intersex issues.[9]

The next post of Intersex Flower followed the same day, and they narrate how they discovered they were intersex during puberty and the lack of scientific knowledge from the medical community regarding their "condition". In the post that followed, on 31 August 2009, they explained that doctors presented to their parents only one option, which was the performance of IGM, and the doctors did not inform Intersex Flower even though they were 15 years old and could make decisions for themselves. On the same day, Intersex Flower uploaded a new text, which presented in more detail how they were mistreated by several doctors and in different hospitals due to a lack of knowledge and prejudice. At the end of the text, they mention that there will be a new text, but the blog has not been updated since 2009. Still, the first online denouncement of medical violence against intersex people remains on the internet, signalling the public birth of the intersex movement in Greece.

### 2.2. Out and Proud! Standing Up for Intersex Rights (2017–2021)

On 26 October 2017, the day of intersex visibility, Lakis Kandilis gave the first interview as an intersex person to *Antivirus Magazine* (Thanopoulos 2017b). He explained that he chose to openly identify as an intersex person with the intention of increasing the visibility of such individuals (Ibid.). He shared his story saying that in 2010, at the age of 28, he experienced severe pains followed by blood loss. After visiting several doctors who could not issue a diagnosis, he was sent to Thessaloniki Hospital. There, a team of 12 doctors determined that he was born with XXY. However, due to the absence of current scientific knowledge, they terrified him by suggesting that by the age of 35, he might develop breast cancer (Ibid.). Later in 2018, it appears that Kandilis talked publicly during the Radical Pride of Thessaloniki.[10]

Before Kandillis, on 6 October 2017, Irene (Rinio) Simeonidou gave her first interview to the same magazine, a journal primarily addressed to Lesbian, Gay, Bisexual, Trans, and Queer (LGBTQ) people (Thanopoulos 2017a). She mentioned that she became aware of intersex in 2009 when she was pregnant with her second child. The doctor at the local hospital recommended terminating her pregnancy solely because they had identified a karyotype that was "uncommon." Despite the doctor's advice, she decided not to terminate

her pregnancy and to give birth to her child. Afterwards, she established a webpage, and numerous families reached out to her facing similar situations: doctors advising them to terminate pregnancies due to the likelihood of their children being born intersex. She also noted that families who chose not to terminate faced pressure from doctors to undergo surgeries on their children in order to conform to the female–male binary.

Simeonidou's interview was the first to shed light on selective abortions that take place because the foetuses have karyotypes that fall outside of the XX and XY binary. These karyotypes, apart from their divergence from what is considered "normal," do not display any diagnosable illness. This was probably the first interview to spark a dialogue on selective and forced abortions not only at the national but also at the international level. At present, the issue is included in the demands[11] and submissions (See, for example, (Written Individual Submission 2018, p. 2)) of the intersex community and in public speeches[12] of intersex activists, but no other parent[13] has raised it publicly. It is worth noting that even though in 2022 the United Nations Committee on Economic, Social and Cultural Rights in its General Comment No. 22 characterised selective abortions as a form of violence against intersex persons, intersex advocacy still focuses primarily on the performance of IGM as a form of violence and torture.[14] This may be due to two factors. First, there is not as much data available on selective abortions as on the performance of IGM.[15] Second, the topic of abortions may be considered as more "controversial" and including it as a top priority in the intersex advocacy agendas could lead to a backlash. Especially during recent years, there have been rapid developments concerning the right to abortion[16] in many countries worldwide, exacerbating the controversies that have surrounded the issue for decades (See Kelley et al. 1993; Tamney et al. 1992).

Following this interview, Simeonidou garnered increased attention. From 2017 onwards, despite not being an intersex individual herself, her story became the primary result on the internet when someone searched for information on intersex. After her appearance in the Greek Parliament in 2017 (See Section 3), she gave an interview to *LiFO* (Antonopoulos 2017), a mainstream and influential publication, where she shared her personal story and parenting experience. She also focused on the alliances she had made, as in the absence of intersex-led organisations, she partnered with national LGBT organisations such as Rainbow School and Rainbow Families (Verde 2017). In 2019, she published an article together with a picture of her child from the European Intersex Community Event (Simeonidou 2019), meaning that she had been receiving support from OII Europe. She focused again on her story and provided some data, stating that "today we know that more than 80% of intersex pregnancies are terminated unnecessarily, with medical prompting or pressure and the frightened consent of uninformed or fearful parents, purely because of prejudice" (Ibid.). The data to which she referred are probably based on information personally gathered by her over the years, as she did not attach any source. The same year, she shared her story during TEDxLesvos (TedxTalks 2018), and at present, this video has had 7.300 views.

Simeonidou's narrative continues to be the most widely covered in Greece. However, it is noteworthy that she is not intersex herself. Her story primarily revolves around selective abortions, setting it apart from other parents' accounts that often centre on decisions related to the performance of Intersex Genital Mutilation (IGM) on their children.[17] The publicity and acceptance that her story received could originate from Greek culture and values, and it could be linked with the role of the mother in Greek society (See Romero and Cid López 2018; Myers n.d.). Davaki states that in Greece, the heteronormative family is considered a sacred institution and motherhood is highly valued (See Davaki 2013). Tsouroufli notes that Greek literature is replete with heroic, selfless mothers who have suffered silently to protect their honour and children (Tsouroufli 2020). Christensen also underlines that, in Ancient Greece, women, and especially mothers, were of decisive importance in the world of poetry and mythology, and there was an entire genre of poetry dedicated to telling the stories of heroic families based on daughters-in-law and mothers, who helped humanise the heroes (Christensen 2023). It could be assumed that Greeks appeared to be receptive

to Irene's story because she impersonates a "selfless", "modern hero" mother who did everything to protect her child's life despite the doctors' advice.

According to the official website of Intersex Greece, the organisation began its activity as a "collective" and online. It started as a small group on Facebook, which, by 2021, had more than 240 members, including 120 families of intersex children and adults.[18] This small group evolved into a nationwide organisation of intersex people, their families, and allies located in all parts of Greece.[19] After the establishment of the organisation, Simeonidou stated that through the provision of peer-to-peer support, "more than 10 children who would have been victims of ignorance or misinformation have been saved in the last decade from involuntary terminations of pregnancies or cosmetic surgeries" (Elmatzoglou 2021). This statement underscores that putting an end to such practices is one of the organisation's foremost priorities. The first public event of the organisation took place in 2021, making it a landmark year for the visibility of the intersex movement in Greece.[20] Drawing from information available on the organisation's Facebook page, in the years that followed, they started raising awareness on intersex issues through events that were addressed to the medical community, teachers, families and the public in general.[21] In 2021, Kandilis called intersex people in the country to join the movement (Antivirus Magazine 2021), but he remains the only intersex person to have come out publicly, and even though Intersex Greece is intersex-led, there are no other members to date that have shared their experiences publicly.[22]

Intersex Greece managed to achieve legal change the year after its official establishment, and this is extremely rapid compared to the time that other organisations required in Greece to achieve similar results. For instance, the Transgender Association was officially established in 2010,[23] and it achieved legal change seven years later, with the law on LGR (See Section 3). This very rapid growth of the organisation could be explained by the size of the global intersex movement compared to other LGBT movements.[24] The global count of intersex activists is relatively modest, but with the backing of ally organisations, intersex groups are actively collaborating, supporting one another, and achieving their advocacy objectives (Howe et al. 2017, p. 23).

### 3. Law 4491/2017: A "Missed Chance" for Intersex Rights?

The first attempt to legally ban IGM in Greece was in 2017. Before 2017, the only country that had legally banned IGM was Malta, with the Gender Identity, Gender Expression and Sex Characteristics (GIGESC) Act.[25] At the time, Malta had elected a Labour-led government, which created an LGBT-friendly Ministry for Civil Liberties. Still, there were no intersex-led organisations in the country, and this gap was filled by international intersex activists. After the IGM ban in Malta, it was reported that governmental delegations, including those from Greece, visited the country to inform changes to their own laws (Howe et al. 2017, p. 9). Indeed, the Maltese law with a focus on the protection of gender identity is referenced several times in reports by the Greek National Commission on Human Rights in 2015[26] and the documents published by the Hellenic Parliament together with Law 4491/2017.[27]

In 2017, the coalition of the Radical Left (SYRIZA) and the Independent Hellenes (ANEL) governed Greece (Mylonas 2018, p. 121). The economy remained fragile, meaning that the Bill was passed in a period of political instability (UNHCR 2017). The Bill was drafted by a committee consisting mainly of academics (Kaiafa-Kmpanti et al. 2017, pp. 1, 7), and it appears that the only member of the civil society involved was the Transgender Association. However, according to its president, Marina Galanou, they did not have the chance to be actively engaged throughout the process (Ibid, p. 7). On 3 April 2017, an event was organised in Thessaloniki where academics, politicians, and activists who took part in the law-making talked about the Bill that was already submitted to the Ministry of Justice on 18 November 2016. In their presentations, they mention intersex several times, and a member of the committee, Professor Katerina Fountedaki, analysed Article 2, which provides the definition of "sex characteristics," and Article 7 (See Appendix A), which bans IGM (Ibid.).

The term "sex characteristics" used in the text is a legal term that was introduced in 2015 by the Maltese Act to offer protection to intersex people for the first time. Since then, the term has been widely used in international and European documents to refer to intersex people (For example, see Free & Equal UN 2017; The European Parliament 2019). Even though Article 2 remained in the text that reached the Parliament, Article 7 disappeared when the text was delivered from the Drafting Committee to the Ministry of Justice, Transparency and Human Rights and never reached the Parliament.[28]

In June 2017, and before the Bill reached the Parliament, SYRIZA organised an event during Pride Festival and the General Secretary of Transparency and Human Rights and said that there would be a new Bill specifically on intersex children in collaboration with the Ministry of Health (Left 2017). Later, in September 2017, Simeonidou went to the Parliament, representing the organisation "Rainbow School" and presented the issues that intersex children face and shared her story on sex-selective abortions. While at the parliament, Kontonis, who was the Minister of Justice, Transparency and Human Rights, told her that "this bill does not refer to intersex persons" (See Appendix A) and repeated what the General Secretary had earlier said about a new Bill on intersex. It is clear that the intentions of the Drafting Committee were to include the ban of IGM in the Bill, while the intentions of the Ministry were to only limit this Bill to trans individuals; hence, Art. 7 was deleted before reaching the parliament. Unfortunately, Art. 2 on "sex characteristics" was not deleted, and this sparked endless confusion and controversial discussions during the voting process. In the Explanatory Memorandum of the Bill, "sex characteristics" is used to refer to trans individuals and elaborate on the fact that the law protects the gender identity of trans persons through their LGR and "sex characteristics" since they will not have to go through a surgical operation to access LGR (Ibid.). In the same document, the term "biological characteristics" is used in an effort to explain that the Bill applies to both trans and intersex people, as both—whether they are trans or "middlesex"—experience "gender dysphoria."[29] On the contrary, the Report for the Bill, which was prepared by the Scientific Service of the Parliament, does not link the Bill to intersex people (Ibid.).

The confusion escalated during the discussions at the Parliament on 9–10 October 2017. Delis, Giokas and Pafilis from the Communist Party of Greece stated that transgender persons experience a conflict between their "sex characteristics" and the gender with which they identify (Ibid.). This is not in line with the definitions issued by human rights institutions. For example, according to the definition of the Commissioner for Human Rights, "transgender persons include persons who have a gender identity which is different from the gender assigned to them at birth" (Commissioner for Human Rights 2011, p. 132). Karakostas, Michaloliakos, Koukoutsis from Golden Dawn referred to sex characteristics to show that they are strictly biological and therefore gender cannot change as it is predetermined during the foetal phase (See Appendix A). In that case, the term "sex characteristics" was used by the far right to rationalise their opposition to the Bill, claiming that they combat the destruction of the human race—a theory used widely by the anti-LGBTI movements (See Strand et al. 2021). Kiriazidis of Nea Dimokratia said that more experts are needed to verify whether an individual is eligible for LGR because the Bill, as it stands, is confusing as it includes "sex characteristics" in Art. 2, which are biological. However, the rest of the provisions stipulate that they can change according to the person's will (See Appendix A). Amidst this confusion, Kontonis clarified that the Bill was not about intersex, and he referred to the developments regarding the protection of their rights at the Council of Europe (CoE): "intersex people should also be given the opportunity to identify themselves and not necessarily in the male-female binary, but that there should be a blank or third entry, something that we must also take seriously into account in this bill and that we will submit the relevant amendment". (Ibid.). Kontonis was referring to Resolution 2191 (2017) (Parliamentary Assembly 2017), but Art. 7 was about the ban on surgeries performed on intersex infants and not about the LGR of intersex. Later, Ourousidis from SYRIZA referred to Simeonidou's presentation and linked surgeries and abortions to LGR

as doctors aim to either fit intersex people in the binary or terminate their lives before they are born due to the absence of their LGR (Ibid.).

It is with no doubt that IGM could have been banned in Greece in 2017, but the Ministry decided that it should be postponed in a political move that is not surprising. The same occurred with civil unions, which used to be exclusionary against same-sex couples, and Law 4356/2015 granted them access when the country was called to implement the ECtHR's decision in Vallianatos and Others v. Greece. Regarding trans rights, before Law 4491/2017, the developments at the ECtHR with Goodwin v. UK and at the CoE with Resolution 2048 (2015) (Parliamentary Assembly 2015) were referenced many times in key events and documents.[30] Nevertheless, the exclusion of intersex people from the Bill was profoundly problematic, revealing a glaring lack of education on intersex issues among the vast majority of those involved in the entire legislative process. The term "sex characteristics", intended to safeguard the rights of intersex individuals, unfortunately, had adverse effects on trans rights and led to the complete invisibility of intersex people. In his first interview, Kandilis stated that Law 4491/2017 was a "missed chance" for intersex rights.[31] Indeed, the legal protection of the rights to bodily integrity and self-determination for intersex infants and children has been established since 2017. However, the effective implementation of Article 7 would likely have faced considerable challenges, given the widespread misinformation about intersex issues and their frequent confusion with transgender concerns.

### 4. Law 4958/2022, Articles 17 to 20: "A Law about Us with Us"[32]

From 2017 to 2022, three additional European countries legally prohibited IGM: Portugal, Germany[33] and Iceland.[34] In Greece, in 2022, Nea Dimokratia was in power, having secured a single-party majority in the 2019 elections (Freedom House 2022). The economy was slowly recovering compared to the conditions in 2017 (Bank of Greece Monetary Policy 2023). On 17 March 2021, the prime minister decided to form a committee to draft the National Strategy for Equality of LGBTQI+ (National Strategy for the Equality of LGBTQI+ 2021). This development followed the European Union's LGBTIQ Equality Strategy 2020–2025 (European Commission 2023). For the first time, the National Strategy incorporated an analysis of intersex issues (National Strategy for the Equality of LGBTQI+ 2021, pp. 51–53), following a comprehensive submission received by the Committee from Intersex Greece (Intersex Greece 2021). This submission proved immensely valuable, marking the first instance where intersex issues were distinctly separated from transgender ones. Finally, the term "sex characteristics" was appropriately employed, and the approved term "intersex"—as recognised by the Greek community—was consistently utilised throughout the text. Moreover, the submission established the primary priorities for the protection of intersex rights, with the foremost priority being the prohibition of "normalising" surgeries on intersex infants and children. Additional priorities encompassed advocating for the inclusion of the protective term "sex characteristics" in all relevant legal documents for intersex people,[35] ensuring access to their medical records and promoting inclusive and non-pathologising healthcare services, especially for intersex infants and children. It also involved creating mechanisms for psychosocial support for intersex individuals and their families, integrating intersex issues into education and awareness-raising programs, establishing a secure school environment for intersex individuals, and introducing integration programs for intersex people in employment.

Shortly after the publication of the Strategy, Intersex Greece began working closely with the Ministry of Health on the draft text of Law 4958/2022 (Intersex Greece 2022), and this is the first main difference compared to the drafting of Law 4491/2017: an intersex-led organisation was involved in the process, whereas previously only academics and a trans-led organisation were engaged. The second substantial difference lies in the legislative context. In 2017, Articles 2 and 7 were part of a Bill related to transgender rights crafted by the Ministry of Justice. In contrast, in 2022, Articles 17 to 20 were incorporated into a Bill focused on reforms in medically assisted reproduction,[36] prepared by the Ministry

of Health. This indicated that the Bill was not directly linked to LGBT rights but rather centred on sexual, reproductive, and women's rights.

Initially, Articles 17–20 were consolidated under Article 16 of the Bill, and they were separated when the Explanatory Memorandum was released. This provision specifically prohibits medical procedures and treatments, including surgical or hormonal interventions, for the total or partial alteration of sex characteristics in intersex minors below the age of 15.[37] The intersex minor could go through such procedures only after permission, which is granted with the decision of the County Court, following the opinion of an interdisciplinary committee. In situations where medical operations are conducted, leading to a misalignment between the gender of the intersex person and the initially registered gender, it is feasible to rectify the latter through a court decision.[38] Physicians who fail to adhere to the stipulations outlined in the provisions are subject to disciplinary and administrative sanctions, a minimum imprisonment term of six months, fines, and, in any case, are barred from practising medicine. The Ministry of Health published an Analysis of the effects of the regulations of the Bill and stated that Art. 16 "protects the bodily integrity of intersex persons and ensures the normal development of their gender and their right to bodily self-determination." (See Appendix B). The Memorandum contained exactly the same information as the report on the analysis of the effects. The Report of the General Accounting Office of the State, which was published on 8 July 2022, added another effect of the Bill that was not highlighted in previous documents—it mentioned that Art. 20 on the sanctions of doctors who perform IGM will possibly increase the public revenue (Ibid.).

The Bill was published for online public consultation from 20 June 2022 to 4 July 2022, and a total of 132 comments were received, of which only 5 concerned the provisions on intersex.[39] All five comments focused mainly on the word "intersex" (ίντερσεξ) and proposed that the Greek word "diafylikos" (διαφυλικός) should be used because this is a national text (Ibid.). The number of comments that this Bill received was significantly lower than the Bill on LGR, which received 863 in total.[40] During the time of the consultation, Intersex Greece's members gave interviews to mainstream media to educate the public using scientific data and stories of families of intersex children (See Pikramenou and Rinio 2022; Maxouri 2022). The Minister of Health had announced as early as May that, following the introduction of the provision banning conversion therapies, the Ministry would also incorporate a provision specifically addressing intersex issues (Iefimerida 2022). It seems that the Ministry deliberately did not include the IGM ban in the Bill on conversion therapies since this could create similar tensions to those in 2017.

On 13 July 2022, Intersex Greece's members attended the hearing of Civil Society Organisations (CSOs) at the Parliament. Simeonidou shared her story on sex-selective abortions and stories from intersex survivors of IGM practices (Papaioannou 2022). On 19 July 2022, the Bill was voted on, and discussions were predominantly centred on matters related to women's rights. Notably, the new provisions allowed women aged 54 to access Assisted Reproductive Technologies (ART) and cryopreservation for social reasons, generating considerable debate (See Appendix B). Agathopoulou from SYRIZA and Arsenis from DiEM25 had several objections when it came to the safeguarding of women's rights and the commercialisation of ART. Nonetheless, regarding the provisions of IGM, both officials stated that they would vote in favour. Agathopoulou also referred to previous fruitless efforts saying that this was a "longstanding demand of the LGBTQI community, as also articulated in a conclusion by a committee of the Ministry of Health during the SYRIZA government." (Ibid. See also Section 3). A pathologising approach to intersex was adopted by Athanasiou from Greek Solution and Markou from SYRIZA, who were both parliamentarians and physicians. Athanasiou said that, in the case of intersex children, the doctor has to guide the parents during prenatal screening, confirming implicitly the practice of sex-selective abortions. Markou considered the word "intersex" inappropriate and blamed it on the fact that no endocrinologists were involved in the process, and he suggested the use of the pathologising term "congenital malformations". (Ibid.). Apatzidi from DiEM25 raised questions regarding the age limit, the conduct of

operations under medical necessity, and the requirement for court involvement. Lioupis from Nea Dimokratia argued that the age limit was a positive element but did not justify it. Euthumiou from Nea Dimokratia said that these provisions followed the prohibition of conversion therapies, confirming the information mentioned above, namely that the IGM ban was strategically included in this Bill. The Prime Minister and the Minister of Health emphasised a "reliance on emotions" and centred their attention on Simeonidou's speech, expressing how deeply moved they were by the challenges faced by intersex children and their families over the years (Ibid.). This was also a strategy followed by Intersex Greece, as they always shared multiple personal stories along with scientific data to inform the public and inspire empathy.

The provisions on the IGM ban passed almost unanimously, and they were among the only ones that did not receive "NO". Despite the success, it is doubted whether those who voted had a sound understanding of intersex. After all, in the course of the vote, the words "sex change" and "transgender" were sometimes used in an ambiguous manner,[41] and no in-depth information was given regarding the law.[42] Art. 17 of Greek law 4958/2022[43] stipulates that an intersex minor who is older than 15 years old may undergo medical operations and treatments only with their free and informed consent. On the consent of a minor, the provision refers to Law 3418/2005, which, according to sub-paragraph aa), states, "If the patient is a minor, consent shall be given by those who exercise parental care or guardianship. However, their opinion shall also be taken into account if, in the opinion of the doctor, the minor has the age, mental and emotional maturity to understand their state of health, the content of the medical act and the consequences or effects or risks of that act". If the intersex minor is under 15 years of age, medical procedures and treatments are prohibited. However, if permission has been granted by the County Court, medical acts are allowed. The permission is granted only for medical procedures and treatments that cannot be postponed, provided that they will not cause future and irreversible complications to the minor's health. The County Court follows a non-contentious proceeding according to which the court may, without the existence of a pre-existing dispute, grant judicial protection for the purpose of safeguarding or protecting the interest of the intersex minor and therefore, the permission granted by the court is not subject to appeal. The hearing is held behind closed doors to protect the privacy of the intersex child. For permission to be granted, the following are required: (a) an opinion of a multidisciplinary committee; (b) a hearing of the representative of the interdisciplinary committee; and (c) a hearing of the intersex minor by the judge. The permission is not required when the medical procedure or treatment is necessary to prevent a risk to the life or health of the minor within the meaning of Law 3418/2005: "3. In exceptional cases, consent is not required: (a) in urgent cases, in which appropriate consent cannot be obtained and there is an immediate, absolute and urgent need for medical care and (c) where the parents of a minor patient or the relatives of a patient who cannot for any reason consent or other third parties who have the power of consent for the patient refuse to give the necessary consent and there is a need for immediate intervention in order to prevent a risk to the life or health of the patient".

Based on Art. 18, the interdisciplinary committee consists of one doctor with experience in operations on intersex individuals or any interventions of "normalisation" of sex characteristics, or as they are called in the medical community, "Disorders of Sex Development (DSD)" or "Congenital Anomalies". These medical terms are stigmatising and often unjustifiably pathologise intersex bodies, but they were used in the text to clarify the interventions to which the law refers and to avoid confusion with procedures carried out on trans people following their informed consent. Furthermore, the committee consists of one legal expert with expertise in bioethics, one psychologist with expertise in issues that intersex people are experiencing, one social worker with expertise in issues that intersex people are experiencing, and one representative of the intersex civil society with relevant expertise in issues that intersex people are encountering. Art. 19 stipulates that in the case that medical operations are performed and result in a discrepancy between the gender of the intersex person and the already registered gender, there is the possibility of amending

the registered gender by court decision. It is worth noting that, in Greece, there are two genders on birth certificates and identity documents: "female" and "male". Therefore, intersex persons are not yet legally able to identify as they wish if their gender does not comply with the female–male binary. In other countries in Europe, such as Germany and Austria, and in the world, such as Australia, intersex people have the option to self-identify outside the binary using "diverse", "X", or "other". So far, in Greece, there are no similar legal developments.

Lastly, Art. 20 states that "doctors who perform medical procedures or treatments to minor intersex persons in violation of Article 17, in addition to the foreseen disciplinary and administrative sanctions, are punished with a prison sentence of at least six (6) months and a fine. The repeated performance of the act of the first paragraph constitutes an aggravating circumstance. In any case and regardless of the amount of the imposed penalty, the guilty is mandatorily punished with the additional penalty of Article 65 of the Criminal Code (Law 4619/2019, A' 95), on the prohibition to practice the profession".

To grasp the Greek legal text, it is essential to realise that it is a combination of the previous Art. 7, which never reached the Parliament in 2017 (See Appendix A), and elements from the Maltese,[44] German,[45] and Icelandic law.[46] In detail, Art. 7 discusses Maltese law, German law, and Icelandic law, which include multidisciplinary committees that, under different circumstances, assess the situation of the intersex minor. Maltese law punishes physicians who perform IGM; German law requires approval by the family court regarding some interventions; Art. 7 and Icelandic law set age limits; and Art. 7 outlined a procedure for the correction of the registered gender of an intersex child who had undergone surgery.

In general, Greek law exhibits two primary strengths. First, it clarifies the terminology by explicitly using the terms "intersex" and "sex characteristics," thereby addressing previous confusion. Second, the legislation strengthens protection by criminalising all practices of Intersex Genital Mutilation (IGM) through sanctions imposed on physicians. The main negative points of the law include the age limit, the presence of a committee, and the court procedure. First, the age limit was never justified in the parliament, even though the question was posed during the vote. The previous Art. 7, which was removed from the LGR Bill in 2017, did not specify an age limit. Moreover, the current law mentions Law 3418/2005 and sub-paragraph aa, which does not indicate a specific age limit either but mentions the doctor as the person responsible for judging whether the minor has the required "age, mental and emotional maturity". Such provision could be proved problematic and therefore it should be crucial to ensure that the minor intersex person over 15 years of age can give informed consent only if (a) the information provided to the minor is based on up-to-date medical information on the risks, medium- and long-term consequences, the availability of alternative medical options, and non-medical information on the living conditions of persons with natural variations of sex characteristics; (b) the minor should be provided with individualised psychological or psychosocial counselling and peer counselling, as it is important that an independent professional with experience in intersex issues (e.g., a psychologist) is involved in the process, in addition to the doctor in charge of the planned intervention or treatment, to assess the minor's ability to consent. As of now, it appears that there is no consensus on those age limits since, for instance, Icelandic law establishes a different limit of 16 years of age. This implies that, in the absence of a consensus, the effectiveness of such age limits will become apparent during the implementation process. Secondly, the presence of a committee might prove complex since, in many cases, a significant portion of its members belong to the medical community. This composition raises concerns about the potential pathologising character of decisions made by the committee. Nonetheless, this is the first law where the committee also includes a representative of intersex civil society. Third, the court procedure, even though it is non-contentious, might also prove problematic due to the lack of training of judges on LGBTI issues in the country; the first—and so far the only—seminar for judges on intersex was held in 2022 (Pikramenou 2022a). OII Europe has highlighted some additional omissions in

the law, which encompass ensuring that mature individuals have access to all necessary comprehensive information for fully informed consent; involving an independent third party to assess a minor's capacity to provide informed consent; recognising the right to psychological and psychosocial support; acknowledging past harm (albeit partially during the voting process[47]); providing for low-threshold means of reparation; and establishing a monitoring mechanism to assess the implementation of the law (OII Europe 2022).[48]

## 5. Conclusions

In 2010, Greenberg said that feminists can benefit from the intersex movement and the broader LGBT movement through the examination of their strategies and alliances (Greenberg et al. 2010, p. 14). In the context of Greece, the intersex movement is inherently feminist. Its origins trace back to the organisation of intersex women, families of intersex girls, and a group of mothers of intersex children. This collective effort laid the groundwork for the formal establishment of the organisation Intersex Greece. Additionally, at the very core of the organisation's demands, there is not only the ban on IGM but also the ban on sex-selective abortions of intersex foetuses, which prevail in Greece, as the parliamentarians implicitly confirmed during the voting process. Even though the organisation has raised the issue multiple times, the global intersex movement and jurisdictions still seem hesitant to set it as a priority.

The case of Greece highlights that the legal prohibition of Intersex Genital Mutilation (IGM) materialised only when an intersex-led organisation actively participated in the process. The primary obstacle to earlier efforts to ban IGM stemmed from a lack of education on intersex issues. At the time, those involved in the process, including LGBT CSOs, lacked a comprehensive understanding of intersex issues. In the absence of an intersex-led organisation, awareness-raising initiatives on intersex were non-existent. Furthermore, terms like "sex characteristics" were misused to advance anti-LGBT ideologies and pathologise both trans and intersex individuals. Later, in 2022, the results of the involvement of Intersex Greece in the process became evident as human-rights-based terminology was used, and the law itself included unique elements, such as the participation of a member of the intersex CSO in the interdisciplinary committee. Moreover, the inclusion of provisions on intersex in the law was facilitated by a less tumultuous political and economic environment compared to 2017. Distancing intersex issues from LGBT concerns, as demonstrated by the challenges faced during the Law on Legal Gender Recognition (LGR), proved to be a strategic move. Moreover, the intersex movement employed the media strategically, not only to inform the public but also to foster empathy, ultimately seeking to overturn the negative image associated with intersex in previous years.

While Greek law is not without its flaws, it does boast some robust elements, making it one of the notable examples among the limited number of laws in existence. The continuous efforts of the intersex movement since the 1990s to ban Intersex Genital Mutilation (IGM) have encountered challenges. Many jurisdictions remain hesitant to challenge the female–male binary, and when they do, it is often approached experimentally, given the lack of consensus on how to effectively ban IGM while safeguarding the rights of intersex individuals. Presently, there is a lack of official governmental data on the implementation of IGM laws due to the absence of monitoring mechanisms. The only available—albeit unofficial—data pertain to Malta and indicate that the law is not fully implemented (See StopIGM 2019; Garland and Travis 2022). Intersex Greece has already expressed its concerns regarding the implementation of the law, citing the very low levels of awareness on intersex issues and the absence of a monitoring mechanism as significant concerns (Intersex Greece 2023). Indeed, the effectiveness of Greek law may hinge on the actions taken in the coming years. Still, the Greek case has already left a valuable legacy, exemplified by intersex activists who have transitioned into law-makers, actively participating in legislative processes aimed at safeguarding their own rights.

**Funding:** This research was funded by the European Union (ERC Consolidator, Abortion Figurations, 101044421).

**Institutional Review Board Statement:** Not applicable.

**Informed Consent Statement:** Not applicable.

**Data Availability Statement:** The original contributions presented in the study are included in the article/supplementary material, further inquiries can be directed to the corresponding author/s.

**Acknowledgments:** In this inaugural publication, I assume the dual role of academic and activist. My heartfelt gratitude extends first to Irene Simeonidou as her invitation to join the intersex movement in 2019 as a legal expert and ally has been transformative. I am deeply indebted to Lakis Kandilis, Vasso Vouvaki, and Eleni Pateraki for their unwavering support and immense trust over the years, alongside all the members of Intersex Greece. This work would have been inconceivable without you. In navigating the law-making process, my sincere thanks go to Dan Christian Ghattas and all members of OII Europe for their invaluable guidance. I express appreciation to Kimberly Zieselman and InterAct for their steadfast support, and I extend gratitude to Alexis Patelis and Prodromos Pyrros for their seamless collaboration. Lastly, I would like to acknowledge Professor Marta Bucholc for her insightful feedback and ethical approach to intersex research.

**Conflicts of Interest:** The views and opinions expressed are those of the author(s) only and do not necessarily reflect those of the European Union or the European Research Council. Neither the European Union nor the granting authority can be held responsible for them. The author declares no conflict of interest.

**Appendix A**

1.   <u>LGR Bill as of 18.11.2016</u>

Article 2. Definitions.

2.   Sex characteristics are the chromosomal, genetic and anatomical characteristics of the individual, which include primary characteristics, such as reproductive organs, and secondary characteristics, such as muscle mass, breast development or hair growth.

Article 7. Minors.

1.   Any medical treatment, such as surgical or hormonal treatment, for the total or partial change of the sex characteristics of a minor is prohibited, unless it is in the best interests of the minor's health, in which case it is carried out with the consent of their parents or commissioner or without consent if the conditions set out in Article 12 para. 3a and c of Law 3481/2005 apply. In this case, prior approval of a Special Interdisciplinary Committee consisting of a paediatric endocrinologist, a geneticist, a paediatric surgeon, a paediatric urologist or paediatric gynaecologist, a paediatric psychiatrist or paediatric psychologist, a social worker and a paediatrician is required for the performance of the medical operations concerned. The manner in which this Committee is to be set up and its operation shall be determined by a decision of the Minister of Health, which shall be published in the Government Gazette. The same medical operations on a minor who has reached the age of 12 shall require the minor's personal consent.

2.   In the case of the performance of the medical operations referred to in par. 1, which results in a discrepancy with the registered gender of the minor, the correction of their registered gender shall be decided by the court, if requested by their parents or their commissioner or the public prosecutor or even ex officio. The court shall adjudicate in accordance with the procedure of voluntary jurisdiction in accordance with Article 782 of the Code of Civil Procedure. The court, depending on the maturity of the minor, must also hear the minor's own opinion as well as the opinion of their parents and decide in the best interests of the minor. The application shall state the new gender, the name chosen and the adjusted surname in relation to it. The application shall be accompanied with copies of the birth and naming certificates of the minor, as well as a copy of the approval of the Special Interdisciplinary Committee referred to in paragraph 1.

2. LGR Bill as it reached the Parliament and then passed into Law 4491/2017

Article 2. Definitions.

Same.

Article 7. Other provisions.

1.  In the first subparagraph of par. 1: "In particular in the case of gender correction, a court decision is sufficient if it is final."

2.  In paragraph. 1 of Article 1 and in paragraphs 1 and 1. 1 of Article 2 of Law 927/1929 (A'139), after the words 'gender identity card', the words 'gender characteristics' shall be added.

3. Explanatory Memorandum of the Bill "Legal recognition of gender identity—National Mechanism for the Development, Monitoring and Evaluation of Action Plans for the Rights of the Child"

Article 1 of the bill contains two general declarations. The first relates to the possibility of trans persons to correct their registered gender and the second to their actual status, as determined by their sex characteristics. It is thus established in the first paragraph that a person has the right to recognition of their gender identity as an element of his or her personality, and in the second paragraph that a person has the right to respect for their personality on the basis of their sex characteristics. (. . .) In particular, the regard for sex characteristics is underscored by the combined effect of Articles 2 and 3 of the Bill, i.e., the assumption that medical interventions for the total or partial change of sex characteristics must be freely chosen by the person concerned and do not constitute a compulsory condition for the person to proceed to legal gender correction.

Article 2 provides definitions of gender identity and sex characteristics. Gender identity is defined in the first paragraph as the internal and personal way in which a person experiences their own gender, irrespective of the gender assigned at birth on the basis of biological characteristics. This way may be in complete contradiction with the person's biological characteristics and the associated assigned sex, but it may also be in partial contradiction if there are mixed biological characteristics, i.e., characteristics that do not fall within the standard definitions of male and female, in which case it would be an 'intersex' person (in the obsolete and inappropriate terminology of 'hermaphrodite'). In other words, the reference in the bill to 'biological characteristics' in general means that the bill applies to—and recognition of gender identity can be requested by—all persons who feel 'gender dysphoria', whether they are transgender or middlesex.

3.   Parliament—Minutes of proceedings

Wednesday 27 September 2017

Stavros Kontonis (Minister of Justice, Transparency and Human Rights): But this bill is not about intersex people.

Irene Simeonidou (Member of the "Rainbow School" group): Yes, but it is about the recognition of gender identity. Intersex people usually have different gender identities, so they are the first in line to use it.

Stavros Kontonis: This will be dealt in a bill that will be submitted jointly by the Ministry of Justice and the Ministry of Health.

(. . .)

Monday 9 October 2017

Page 325

Evagelos Karakostas (Golden dawn): (. . .) The Emeritus Professor of Neurology of the Aristotle University of Thessaloniki stressed that the characteristics of the sex of a person are predetermined, apart from the genome, in the brain, that there are indeed many differences between the two sexes, but the most important difference, which begins from intrauterine life, from the moment of organogenesis of the brain in the fourth foetal month, is in the so-called "amygdaloid nucleus." In conclusion it was said that from birth to death these differences remain unchanged and unaltered. (. . .)

pp. 328–29

Ioannis Delis (Communist Party of Greece): (. . .) If the government wanted to address existing issues that transgender people and middlesex children are facing, then it would adopt the scientifically documented and deeply humane position of the Communist Party of Greece for their full protection and for the effective safeguarding of their rights.

What is this position? The Communist Party of Greece recognises the right of a transgender person to change their gender on legal documents. That is, it recognises it for those cases in which, for biological, social or other reasons, a person experiences an intense internal conflict between the characteristics of their sex and the gender to which they feel they belong to. (. . .) These are theories that distract in an absolute and unscientific way from the biological characteristics of sex, which are, of course, objectively determined by the individual's perception of their gender.

Page 333

Stavros Kontonis (Minister of Justice, Transparency and Human Rights): By the way, this week is the Parliamentary Assembly of the Council of Europe. Normally I should be there now, as should the rapporteur of SYRIZA too, but I stayed in Athens for this particular bill. In fact, on Thursday, the Council of Europe is discussing—and it looks like it will be voted through, since it was passed unanimously by the Committee—the resolution on the elimination of discrimination against intersex people, (in Greek—diafylika) or middlesex people, that is, people born with sexual characteristics that are not exclusively male or female.

It is proposed, on the basis of the resolution, that intersex people should also be given the opportunity to identify themselves and not necessarily in the male-female binary, but that there should be a blank or third entry, something that we must also take seriously into account in this bill and that we will submit the relevant amendment.

Page 349

Dimitrios Kiriazidis (Nea Dimokratia): While you are telling us that a person has the right to respect for their personality on the basis of their sex characteristics, which, of course, according to common experience and logic, are purely biological and relate to the person's physical condition, as you mention in paragraph 2 of Article 2, you then come along and tell us that these characteristics change according to the person's will. But is this a question of will or of reality? (. . .)

Page 356

Nikolaos Michaloliakos (Golden Dawn): (. . .) It is characteristic of what the bill says that gender identity means the internal and personal way in which a person experiences their own gender regardless of the gender from which they were registered after birth based on their biological characteristics. Then, biological characteristics do not matter at all.

Page 383

Georgios Oursouzidis (SYRIZA): The representative of the group "Rainbow School," a parent, said (. . .) In particular, middlesex infants, when their sex image does not fall within the typical image of male-female, the typical external anatomy, their gender is arbitrarily registered. Often this is accompanied by cosmetic and irreversible surgical procedures (. . .) In many cases, in fact, if prenatal testing has been done, parents are advised to terminate the healthy middlesex embryo because there is no provision for the existence of a human being outside the male-female dipole, and as a result healthy middlesex embryos are not even given the right to life.

Page 385

Ioannis Giokas (Communist Party of Greece):For a number of reasons, biological, social or otherwise, (the person is) experiencing an intense conflict between sex characteristics and the gender to which they feel they belong, it is required to be based on certain objective criteria (. . .) and not on a simple application without any social support and protection.

Tuesday 10 October 2017
Page 410

Dimitrios Koukoutsis (Golden Dawn): (...) But—no matter how much you may wish it—biological characteristics cannot be changed. Is it ever possible to give variation and fluidity to the human sex?

Page 414
Athanasios Pafilis (Communist Party of Greece): (...) The Communist Party of Greece recognises the right to change gender in legal documents in cases where the individual experiences a strong conflict between the characteristics of their sex and the gender to which they feel they belong for biological, social, and other reasons.

**Appendix B**

1.  Analysis of the Effects of the Regulation, Title of the Regulation under Assessment: Bill of the Ministry of Health entitled "Reforms in medically assisted reproduction"

Article 16: Intersex persons are persons born with sex characteristics that do not fit the medical or social norms of female or male bodies. These variations may occur in primary sex characteristics (such as internal and external genitalia and chromosomal and hormonal structure) and/or secondary sex characteristics (such as muscle mass, hair distribution and stature). The regulation protects the bodily integrity of intersex persons and ensures the normal development of their gender and their right to bodily self-determination. (...)

2.  Report of the General Accounting Office of the State (Art. 75 par. 1 of the Constitution)

V. On the state budget
Possible increase in revenue from the collection of fines and the conversion of prison sentences into financial penalties in cases of violation of relevant regulations, for the protection of the individual rights of intersex persons. (Article 20).

3.  Parliament—Minutes of proceedings

Eirini—Eleni Agathopoulou (SYRIZA): So, women from fifty-two to fifty-four years old have to something lose. Which women and how many are they? It would be good if you could tell us.
Foteini Arampatzi (Nea Dimokratia): Even if there is just one, why do you mind?
President (Nikitas Kaklamanis): What are we doing now?
Eirini—Eleni Agathopoulou (Nea Dimokratia): Mr President, I see a disturbance that is not justified.
(...)
Finally, with regard to Articles 17–20 on the change of sex characteristics of intersex minors, (...) The regulation is a longstanding demand of the LGBTQI community, as also articulated in a conclusion by a committee of the Ministry of Health during the SYRIZA government and the Transgender Association on the access of LGBTQI people to the health system. The regulation is also welcomed by the non-governmental organisation "Intersex Greece."
Our sole objection in this regard is the potential bypassing of the procedure outlined in the Bill if the situation is deemed urgent, without specifying any procedure or safeguard to assess whether a case genuinely qualifies as urgent. Thus, while the Bill correctly provides for a decision by a magistrates' court, following a recommendation by a special multidisciplinary committee, to operate on a minor under the age of fifteen, the procedure is bypassed if the medical operation or treatment is necessary to prevent danger to the life or health of the minor, without providing otherwise.

Georgios Lamproulis (Communist Party of Greece): The provisions of the proposed bill do not ensure free, all-round, all-systemic, social support for intersex people (...) Specifically, interdisciplinary support should be based on the cooperation of paediatricians, endocrinologists, urologists, surgeons, general pathologists, paedopsychiatrists, psychologists and social workers with appropriate specialisation.
(...) Corresponding scientific, social support, of course, is needed throughout the life course of an intersex person, even after adulthood. The opinion of the scientific committee is

therefore necessary even after the age of fifteen, in order to ensure the safeguard of the protection of the rights of children. Of course, here too, the Ministry, the Minister, did not present the corresponding scientific data.

Maria Athanasiou (Greek Solution): With regard to intersex children in the third part of the Bill, articles 17–20, we refer to the fact that intersex children with down syndrome, turner syndrome, etc., are born this way because of a mistake in the reduction process. None of the parents are to blame, it happened. However, the doctor has to guide in the framework of prenatal screening. After all, this is the purpose of the check-up. In any case, intersex children are born sterile and do not become sterile after the operation, because we also heard this in the committee. (. . .)

Maria Apatzidi (DiEM25): (. . .) The articles for us are moving to a positive direction, of course according to the collective Intersex Greece that participated in the consultation of the bill. (. . .) How can the requirement of a court order for medical procedures for intersex children under fifteen but not from fifteen to seventeen be justified? (. . .) Furthermore, how can a distinction be justified between, on the one hand, medical operations relating to the sex of intersex children and, on the other hand, other operations which are considered necessary for the survival of the individual or to ensure their mental and physical health? For example, routine surgeries, transplants, and so on. (. . .) Why is court intervention required in order to correct the registered gender of the intersex child?

Athanasios Lioupis (Nea Dimokratia): An age distinction is made. It is recognised that those persons who wish to do so and are older than fifteen years of age, provided that parental consent is given, are allowed to undergo medical procedures and treatments (. . .) An attempt is therefore made to respect the wishes of intersex minors and, concurrently, to improve their social and economic situation. The representative of the Association Intersex Greece said that she was delighted with the new regulations. (. . .)

Konstantinos Markou (SYRIZA): (. . .) And I think it is inappropriate for a bill, in a Government Gazette, to say "intersex." It is wrong. So you can—I inform you and as an endocrinologist (. . .) And as a specialist, well, I say that you can adopt the term "congenital malformations," which describes exactly the group you want to support. Of course, you do not have endocrinologists in the committee, which goes without saying (. . .). I will end by saying that you had the opportunity with this bill to improve a little bit the tragic situations of transgender people who have sex change operations, mastectomies, hysterectomies. They are not covered. (. . .)

Kriton-Ilias Arsenis (DiEM25): DiEM25 will stand against this unprecedented abuse of society. We vote against this bill. We will vote in favour of any positive amendments for intersex people and HIV-positive people, but we will never participate in this mill of favours that you have set.

Anna Euthimiou (Nea Dimokratia): I believe that today is a historic moment for all intersex children in Greece with these provisions (. . .) They follow the provisions of the prohibition of conversion therapies, which you, Minister, and I was the rapporteur on the Personal Doctor Bill, introduced.

Kiriakos Mitsotakis (Prime Minister): (. . .) Mr. Minister, I listened carefully, and I am not hiding my emotion, to what the representatives of intersex people testified in the committee of the Parliament, and I learned a lot that I did not know about what happens on the fringes of Greek families, without most of the time being widely known. I was sincerely sorry for the mistakes of the past that led to dramas because we lacked knowledge and courage, and I realised how important the initiatives we are taking today are for these fellow citizens (. . .)

Yianis Varoufakis (DiEM25): Yes, we have to agree and congratulate you, Minister, on the sex change regulations for the LGBTI community, for intersex people.

Athanasios Plevris (Minister of Health): (. . .) I honestly felt embarrassed when I heard the stories of these people and I think everyone on the committee was moved when we

heard the representative of their organisation. (. . .) A concern was raised by the special rapporteur for DiEM25 because there is always this, there is always this in all the processes of medical operations on a minor, that there may be a moment when the doctor has to do something quickly. It is an emergency situation that exists and is foreseen everywhere. For these scenarios, we have established a more stringent framework compared to the framework governing other medical procedures you mentioned. That is, if the child requires participation in a clinical drug trial, a transplant, or a life-saving surgery. This framework is precisely more stringent because we consider that these are interventions that are not linked at that moment to the person's life and the risk but are linked to a decision that will accompany them for the rest of their lives. So we consider it to be a weighted provision which was of course also supported by the Intersex Greece community.

Konstantina (Nantia) Giannakopoulou (PASOK): I will conclude with the very important article, which has to do with the provisions concerning intersex minors, where for the first time they are given the opportunity to undergo medical operations and sex change when they reach the age of fifteen only with their free consent, after informing themselves or persons exercising parental guardianship.

## Notes

[1]  (Free & Equal UN 2017), What does "intersex" mean?

[2]  Ibid. It is essential to note that this information serves to provide the reader with a general understanding of intersex statistics; however, from a non-discrimination and human rights perspective, numbers do not really matter when explaining who intersex people are. See also (Pikramenou 2019).

[3]  In this article, the term Intersex Genital Mutilation (IGM) is preferred as it is also used by the organisation Intersex Greece on their official website—see https://intersexgreece.org.gr/intersex-101/. accessed 22 March 2023. It should be noted that there is research criticising this term; for instance, Rubin, in the article (Rubin 2015), shows how the language of IGM is based on an analogy with Female Genital Mutilation (FGM) that could potentially have imperialistic implications.

[4]  See, for example, Monro et al. (2021), where a variety of novel insights on intersex is offered. However, data originate mostly from regions in Europe, such as Western and Northern Europe.

[5]  See OII Europe, About OII Europe, https://www.oiieurope.org/about/, accessed on 22 March 2023.

[6]  A comprehensive list of intersex groups can be found here: (InterAct 2022), last updated on 7 November 2022.

[7]  Intersex Flower Greece (2009). The pronoun "they" will be used when referring to Intersex Flower in the text as their preferred pronoun is not mentioned in the blog.

[8]  In July 2023, the ECtHR found a violation of Art. 14 together with Art. 8 for discrimination on grounds of sex and sex characteristics: (Judgment 2023).

[9]  The internet has consistently played a pivotal role in intersex activism, inspiring numerous activists to openly express their identity and advocate for their rights. For example, Irene Kuzemko has stated that thanks to the online public intersex figures, she realised that her story was not unique (Kuzemko 2023).

[10]  Intersex Greece (2021). See also Section 32, The Establishment of Intersex Greece (2021–2022).

[11]  In the (Malta Declaration 2013), intersex activists stated explicitly "to put an end to preimplantation genetic diagnosis, prenatal screening and treatment, and selective abortion of intersex foetuses".

[12]  See the keynote lecture by (Cabral Grinspan 2020, p. 2).

[13]  In 2023, Simeonidou shared her story at the Council of Europe at the Conference Advancing the Human Rights of Intersex People (2023).

[14]  For an extensive list, see (Ghattas 2019), Appendix https://www.ilga-europe.org/files/uploads/2022/04/Protecting_intersex_in_Europe_Appendix.pdf., accessed on 23 March 2023.

[15]  The most comprehensive quantitative study to date seems to be the "World Atlas of Birth Defects" by the World Health Organization (WHO), which was referenced in 2014 by StopIGM.org. in the online article "Selective Intersex Abortions: XXY 74%, Indeterminate Sex 47%, Hypospadias 2%". Such terminations appear to be higher when the foetus is XXY and of indeterminate sex, compared to hypospadias (StopIGM.org 2014).

[16]  For the developments in Europe, see (Bucholc 2022), and for the developments in the United States, see (Coen-Sanchez et al. 2022). Also see (Bucholc forthcoming).

[17]  For example, a parent from Iceland shared for the publication #*MyIntersexStory*: "I as a parent made choices that should not have been mine to make and most definitely not the doctors to make, I firmly believe that the intersex individual should be the only one allowed to make choices regarding their own body, there must be an end to unnecessary medical treatment and

surgery of intersex individuals without their consent. We as parents should not have the right to give this consent" OII Europe, #MyIntersexStory, p. 59, 2019 https://oiieurope.org/wp-content/uploads/2019/11/testimonial_broch_21-21cm_for_web.pdf, accessed on 5 April 2023. See also (Audr XY 2019).

18    Intersex Greece, About us, https://intersexgreece.org.gr/en/about-us/#ld-1618924549019-e7e5159b-e5d8, accessed on 6 April 2023.

19    "With the decision no. 477/2021 of the Athens County Court, the statute of the association with the name" Intersex Greece-Greek community of Intersex" was approved" from Intersex Greece's Facebook page, https://www.facebook.com/IntersexGr/photos/a.108099634748969/255653043326960/, accessed on 20 April 2023.

20    Intersex Greece, 1st Public Event for Intersex Human Rights in Greece—"Intersex Rights and Claims in Greece", https://intersexgreece.org.gr/en/2021/06/09/1st-public-event-for-intersex-human-rights-in-greece-intersex-rights-and-claims-in-greece/, accessed on 10 April 2023.

21    See, for example, on the Facebook page of the organisation "Ελληνική Κοινότητα ίντερσεξ—Intersex Greece": Event discussion on the topic "You are born intersex, you do not become one" in the city of Xanthi on 4 February 2022 at 19:00 at Filoistron Café, https://www.facebook.com/IntersexGr/photos/a.108099634748969/281958234029774/, accessed on 20 April 2023. Presentation of the outcomes of the research "Hate Speech against intersex people in Greece", Public Market of Kipseli, Athens on 15 September 2022 at 17:00, https://www.facebook.com/events/625178702648699/?ref=newsfeed, accessed on 20 April 2023.

22    The same issue has been highlighted by other organisations in neighbouring countries; see (XY Spectrum 2018, 0′50″).

23    The Transgender Association's homepage (in Greek), https://tgender.gr/ accessed on 24 April 2023.

24    For LGBT movements and legal change, see (Barclay et al. 2009).

25    For more information on the GIGESC Act, see Ministry for Social Dialogue, Consumer Affairs and Civil Liberties, GIGESC Act, https://meae.gov.mt/en/public_consultations/msdc/pages/consultations/gigesc.aspx., accessed on 25 April 2023.

26    See (National Commission on Human Rights 2015). It has to be noted that the terminology used in this report is not accurate as it uses "middlesex" (μεσοφυλικό) to describe intersex and "diaphyliko" (διαφυλικό) to describe trans persons, even though it is the Greek term for intersex.

27    See Explanatory Memorandum of the Bill "Legal recognition of gender identity—National Mechanism for the Development, Monitoring and Evaluation of Action Plans for the Rights of the Child, https://www.hellenicparliament.gr/, accessed on 24 April 2023.

28    See (Pikramenou 2019, sct. 4.3.3.). "Greece", 2019.

29    Ibid. Note that both the terms "gender dysphoria" and "middlesex" are non-human-rights-based terms that stigmatise and pathologise the trans and intersex communities.

30    See, for example, (Kaiafa-Kmpanti et al. 2017) and the Report for the Bill prepared by the Scientific Service of the Parliament, 2017 (both in Greek).

31    See Section 2 above on the establishment of Intersex Greece.

32    The title is inspired by Intersex Greece's slogan "Nothing about us, Without us".

33    For a comprehensive overview of intersex laws in Europe until 2019, see (Pikramenou 2019), Chapter 4: Intersex in Europe. For a more recent overview, see (Mestre 2022).

34    For an analysis of the Icelandic law, see (Alaattinoglu 2022).

35    Until now, "sex characteristics" have been included in some legal documents, such as Law No. 4443/2016, since one of the grounds for τηε protection against discrimination in the field of employment. Law 927/1979 was amended with Par. 2 of Article 7 of Law 4491/2017 and added "sex characteristics" to public incitement to violence or hatred. Law No. 4619/2019 amended the Penal Code and Article 82A on crimes with racist characteristics and added "sex characteristics" to the list of aggravating circumstances. Law 5029/2023 "We live together in harmony—Breaking the silence": regulations for the prevention and treatment of violence and bullying in schools and other provisions included sex characteristics in the actions of the Ministry of Education on bullying and discrimination in schools (Intersex Greece 2023, pp. 15–16).

36    Bill of the Ministry of Health "Reforms in medically assisted reproduction" (in Greek) https://www.hellenicparliament.gr/en/, accessed on 24 April 2023.

37    The provisions of the Bill, which were afterwards passed into Law, are available here in English: (Pikramenou 2022b).

38    Ibid. Note that, in Greece, there are two genders on birth certificates and identity documents: "female" and "male". Therefore, intersex persons are not yet legally able to identify as they wish if their gender does not comply with the female–male binary.

39    The comments are available (in Greek) in WordPress, www.opengov.gr. Accessed on 24 April 2023.

40    The comments are available (in Greek) at http://www.opengov.gr/ministryofjustice/?p=8074. Accessed on 24 April 2023

41    See Ibid, Varoufakis and Markou.

42    See Ibid., Plevris.

43    See the official website of Intersex Greece: (Pikramenou 2022b).

44    Ministry for Social Dialogue, Consumer Affairs and Civil Liberties, Gender Identity, Gender Expression and Sex Characteristics Act and OII Europe, Press Release: OII Europe applauds Malta's Gender Identity, Gender Expression and Sex Characteristics

Act, 2015, https://www.oiieurope.org/press-release-oii-europe-applauds-maltas-gender-identity-gender-expression-and-sex-characteristics-act/, accessed on 10 May 2023.

45 Deutscher Bundestag, Drucksache 19/27929 (in German) and OII Europe, A good first step: Germany adopts law banning IGM. But there is still room for improvement, 2021, https://www.oiieurope.org/a-good-first-step-germany-adopts-law-banning-igm/, accessed on 10 May 2023.

46 Act on Gender Autonomy No 80/2019 as amended by Act No. 159/2019, No. 152/2020 and No. 154/2020.

47 See Appendix B, Plevris and Mitsotakis.

48 OII Europe, Good practice map 2022 and the 14 IGM ban indicators published by OII Europe in 2023.

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
