# Peer review of "From Intersex Activism to Law-Making—The Legal Ban of Intersex Genital Mutilation (IGM) in Greece"

_socsci, doi:10.3390/socsci13040221_

Round 1
Reviewer 1 Report
Comments and Suggestions for Authors
The article is original and must be accepted with minor changes. The changes the manuscript needs:
- Use acronyms correctly, they should only appear the first time next to their explanation and then only use the acronyms. For example, when talking about Intersex Genital Mutilations (IGM).
- It is necessary to expand the information on the methodology, it is not specified how the study has been carried out: what number of web pages have been included, what inclusion criteria have been used to select them, it is necessary to specify how it has been carried out the search, what search engine was used and what terms were included in the search.
Author Response
1. Use acronyms correctly, they should only appear the first time next to
their explanation and then only use the acronyms. For example, when
talking about Intersex Genital Mutilations (IGM).
The text is already edited by a professional. Acronyms are repeated in
their full form in each section because readers sometimes do not read the
whole text but separate sections instead.
2. -It is necessary to expand the information on the methodology, it is not
specified how the study has been carried out: what number of web
pages have been included, what inclusion criteria have been used to
select them, it is necessary to specify how it has been carried out the
search, what search.
All comments were addressed.
Reviewer 2 Report
Comments and Suggestions for Authors
This article makes a strong and original contribution to the growing field of critical intersex studies. The author traces the history of the intersex movement in Greece, which in 2022 became the 5th country in the world to ban "Intersex Genital Mutilation." The manuscript provides a very detailed history of how intense activists became lawmakers to protect intersex human rights. I believe that the article could be improved by the author engaging with transnational critiques of the use of the term "IGM." For instance, in their 2015 article "Provincializing Intersex," Rubin shows how the language of IGM is based on an analogy with "FGM" that has potentially problematic imperialistic implications. By addressing this critique and showing what the Greek example adds to debates about the legal regulation of intersex medical treatments, the author can strengthen their argument in key ways.
Comments on the Quality of English LanguageThe quality of English language is very strong.
Author Response
I believe that the article could be improved by the author engaging with
transnational critiques of the use of the term "IGM." For instance, in
their 2015 article "Provincializing Intersex," Rubin shows how the
language of IGM is based on an analogy with "FGM" that has
potentially problematic imperialistic implications. By addressing this
critique and showing what the Greek example adds to debates about
the legal regulation of intersex medical treatments, the author can
strengthen their argument in key ways.
This comment was included in the text. The term “IGM” was preferred
because it is also used by the movement in Greece. Rubin’s criticism was
incorporated.
Reviewer 3 Report
Comments and Suggestions for Authors
In general, I think that this article is very original, since it tells the story about the establishment of the recent Greek ban on IGMs. I have not read anything else on the topic, which is in itself an argument for publication.
The author obviously knows very much about the topic, and the account of the establishment process is very detailed. Sometimes the narrative is slightly impeded by the amount of details in the text, which is something that I think the author could keep in mind when revising the text (to always connect the details to the general argument of the article).
The article is very detailed in its description of the discussions and the processes preceding the establishment of the law, but when it comes to the actual law, it just passes over the details quickly. This is a slightly strange choice by the author and does not really give the reader a clear idea of the actual provisions of the law. As a legal scholar, I am missing a clear description of the law and its contents, which would also make the author’s comparisons to other countries’ similar bans clearer. The author’s critical comments would also be easier to understand if the description of the law was clearer.
Some specific comments:
- Numbers of how many people are intersex (p. 1) are always disputed, because they depend on the definition of intersex. The link between the definition and the statistics should be recognised in my opinion, or are numbers really necessary?
- Line 72, I am not sure that the research has focused so much on ‘catholic’ countries as such, perhaps ‘catholic and protestant’ would be closer to the truth?
- The author provides quite compelling arguments for using an ‘online methodology’, but reading the article, I wonder if ethical approval has been given to study and report the contents of private online groups? It would also be important to reflect on the ethical problems in making public discussions which have originally been carried out with the intention of being private. Or are all the posts discussed public already? This was not entirely clear to me as a reader
- ‘mistreated’ on line 122, should this possibly be ‘maltreated’? Or does the author mean ‘mistreated’?
- In Section 2, it would be good to assign every subsection/time period with a time period within brackets, that is not done for Section 2.1 at the moment
- Rather than ‘rapid developments’ on line 163, it may be more appropriate to specify such developments (in other words, restrictions on the right to abortion)
- I am not sure about using the term ‘selective abortions’ when referring to abortions which are carried out on the advice/with pressure from medical authorities, since the situation does not seem to be entirely voluntary in my opinion. I am wondering whether ‘coerced abortions’ would be a better term, since also ‘forced abortions’ may be not entirely appropriate for such situations?
- On pages 9 and 10, it is discussed that IGM was excluded from a 2017 Bill, but the reasons for this remains unclear (just that the Ministry did not want it – but not why), it would be interesting to know more about the reasons behind this exclusion, if possible
- The details of the Greek law could be discussed more in detail. I am not sure, for example, about the criticism regarding existing laws relating to ‘age limits <’(p. 14). The idea is, of course, that people above the age limit (e.g. 16 in the Icelandic case) can decide more freely on interventions on their bodies, while people under the age limit need special procedural safeguards (which is why the law refers to intersex children). Also, when it comes to IGM, this is surely most common at an early age. Age limits do not mean that IGM on older children or adults are permitted, but just that they are able to give informed consent themselves, which children (especially newborns and very young children) are not. Or does the ‘age limit’ mean something else in the Greek case?
- The involvement of a Committee and Court are not clear in the Greek case. In which cases should they be involved, according to the law? In which cases not? What were the motivations behind involving them? I presume that they are considered as procedural safeguards, as IGMs are not fully banned in Greece – is that correct? Are Courts/Committees deciding whether treatments are medically necessary or not? How do these procedures compare to the German legislated procedures, for example? What is the level of necessity necessary? As the article refers to IGMs, I suppose that only genital surgeries are banned/regulated by the Greek law. Is that correct? How about other forms of medical treatment which may not be medically necessary or consensual (e.g. hormone treatment)? These questions should, in my opinion, be clarified in the article.
Comments on the Quality of English LanguageThe article has some typos and syntax errors that I could spot, but the language is generally good.
Author Response
1. The article is very detailed in its description of the discussions and the
processes preceding the establishment of the law, but when it comes to
the actual law, it just passes over the details quickly. This is a slightly
strange choice by the author and does not really give the reader a clear
idea of the actual provisions of the law. As a legal scholar, I am missing
a clear description of the law and its contents, which would also make
the author’s comparisons to other countries’ similar bans clearer. The author’s critical comments would also be easier to understand if the description of the law was clearer.
The comment was addressed and the law is now analysed thoroughly.
2. Numbers of how many people are intersex (p. 1) are always disputed,
because they depend on the definition of intersex. The link between the
definition and the statistics should be recognised in my opinion, or are
numbers really necessary?
Indeed, this comment was addressed.
3. Line 72, I am not sure that the research has focused so much on
‘catholic’ countries as such, perhaps ‘catholic and protestant’ would be
closer to the truth?
Indeed, this comment was addressed.
4. The author provides quite compelling arguments for using an ‘online
methodology’, but reading the article, I wonder if ethical approval has
been given to study and report the contents of private online groups? It
would also be important to reflect on the ethical problems in making
public discussions which have originally been carried out with the
intention of being private. Or are all the posts discussed public
already? This was not entirely clear to me as a reader
All the online sources used are public, there were no sources from private
groups. Each quote has a reference to a link and therefore the readers can
see that these links are public. However, since the author personally
knows the owners of the blogs or interviewees, they are already informed
and have read the exact parts of the text related to them.
5. ‘mistreated’ on line 122, should this possibly be ‘maltreated’? Or does
the author mean ‘mistreated’?
The author means mistreated.
6. In Section 2, it would be good to assign every subsection/time period
with a time period within brackets, that is not done for Section 2.1 at
the moment
Right, comment addressed.
7. Rather than ‘rapid developments’ on line 163, it may be more
appropriate to specify such developments (in other words, restrictions
on the right to abortion)
As indicated in the reference, the author means developments in general not
necessarily restrictions on the right to abortion as abortion was also liberalised
in some countries.
8. I am not sure about using the term ‘selective abortions’ when referring
to abortions which are carried out on the advice/with pressure from
medical authorities, since the situation does not seem to be entirely
voluntary in my opinion. I am wondering whether ‘coerced abortions’
would be a better term, since also ‘forced abortions’ may be not entirely
appropriate for such situations?
The term selective abortions is preferred as it is used in the field and in the
literature that was consulted.
9. - On pages 9 and 10, it is discussed that IGM was excluded from a
2017 Bill, but the reasons for this remains unclear (just that the Ministry
did not want it – but not why), it would be interesting to know more
about the reasons behind this exclusion, if possible
There is no data that show the exact reasons behind it so unfortunately it
is not possible to elaborate more.
10. - The details of the Greek law could be discussed more in detail. I
am not sure, for example, about the criticism regarding existing laws
relating to ‘age limits <’(p. 14). The idea is, of course, that people
above the age limit (e.g. 16 in the Icelandic case) can decide more
freely on interventions on their bodies, while people under the age limit
need special procedural safeguards (which is why the law refers to
intersex children). Also, when it comes to IGM, this is surely most
common at an early age. Age limits do not mean that IGM on older
children or adults are permitted, but just that they are able to give
informed consent themselves, which children (especially newborns and
very young children) are not. Or does the ‘age limit’ mean something
else in the Greek case?
All these comments were answered and included in the analysis of the
law.
11. - The involvement of a Committee and Court are not clear in the
Greek case. In which cases should they be involved, according to the
law? In which cases not? What were the motivations behind involving
them? I presume that they are considered as procedural safeguards,
as IGMs are not fully banned in Greece – is that correct? Are
Courts/Committees deciding whether treatments are medically
necessary or not? How do these procedures compare to the German
legislated procedures, for example? What is the level of necessity
necessary? As the article refers to IGMs, I suppose that only genital
surgeries are banned/regulated by the Greek law. Is that correct? How
about other forms of medical treatment which may not be medically
necessary or consensual (e.g. hormone treatment)? These questions
should, in my opinion, be clarified in the article.
All these comments were answered and included in the analysis of the law.
Thank you in advance,